# Effect of Chitosan Incorporation on the Development of Acrylamide during Maillard Reaction in Fructose–Asparagine Model Solution and the Functional Characteristics of the Resultants

**DOI:** 10.3390/polym14081565

**Published:** 2022-04-12

**Authors:** Hong-Ting Victor Lin, Yen-Shu Ting, Nodali Ndraha, Hsin-I Hsiao, Wen-Chieh Sung

**Affiliations:** 1Department of Food Science, National Taiwan Ocean University, Keelung 202301, Taiwan; hl358@mail.ntou.edu.tw (H.-T.V.L.); ysesst93020@gmail.com (Y.-S.T.); nodali@email.ntou.edu.tw (N.N.); hi.hsiao@email.ntou.edu.tw (H.-I.H.); 2Center of Excellence for the Oceans, National Taiwan Ocean University, Keelung 202301, Taiwan

**Keywords:** acrylamide, chitosan, kinematic viscosity, Maillard reaction products (MRPs), fructose

## Abstract

The objectives of this study were to evaluate the effect of 0.5% chitosan incorporation on acrylamide development in a food model solution containing 0.5% fructose and asparagine after heating for 30 min at 180 °C. All the solutions were investigated for the following characteristics: acrylamide, asparagine, reducing sugar content, color, kinematic viscosity, Maillard reaction products (MRPs), and pH every 10 min. After heating for 10 min, the viscosity of chitosan-containing solutions reduced significantly. The investigational data confirmed that chitosan may have decomposed into lower molecular structures, as demonstrated by the reduced viscosity of the solution at pH < 6 and a decrease in the acrylamide content during 30 min of heating in a fructose–asparagine system. This study also confirms that the formation of ultraviolet-absorbing intermediates and browning intensity of MRPs containing acrylamide prepared by fructose–asparagine was more than those of MRPs prepared by glucose–asparagine solution system. MRPs containing acrylamide resulted from the reaction of asparagine with fructose (ketose) rather than glucose (aldose). Acrylamide formation could be significantly mitigated in the fructose–asparagine–chitosan model system as compared to the fructose–asparagine model system for possible beverage and food application.

## 1. Introduction

Acrylamide and 5-hydroxymethylfurfural (HMF) are harmful food-borne byproducts generated from the Maillard reaction during the heat treatment of starchy foods [1,2,3], such as cereals and bread, as well as oil used in food production, such as French fries and potato chips [2,4]. The possible path in the development of both food mutagens requires Maillard reactions between reactive carbonyl groups of reducing sugar, such as fructose/glucose and amino acid groups, especially asparagine [1,3,5]. The main pathway in the development of acrylamide combines asparagine and reactive carbonyl, resulting in intermediates, such as Schiff and decarboxylated bases, as well as 3-aminopropanamide. The intermediate Schiff base decarboxylates and removes either ammonia or substituted imine to generate acrylamide under heat [6,7]. The first pathway is Schiff base evolution, which occurs after the incorporation of nucleophilic asparagine to the positive carbonyl carbon of the dicarbonyl compound. Asparagine has been identified as a key amino acid that reacts with dicarbonyl compounds and is a precursor in the development of acrylamide as Strecker degradation in the Maillard reaction [7]. The deprived proton from nitrogen and the obtained proton by oxygen were mentioned as the procedure for the reaction by Mottram et al. [8]. In heated model systems [4], 15N-labeled acrylamide can be determined by the rearrangement of glucose and nitrogen-15(amido)-labeled asparagine.

HMF is a heterocyclic compound formed by the Amadori rearrangement of the Maillard reaction in heated carbohydrate and asparagine-rich products, similar to acrylamide [5,9], or by caramelization of the dehydration of hexoses [2].

Chitosan is a partially N-deacetylated derivative and linear chitin extracted from crustacean shells, such as crabs or shrimps [10]. Chitosans have outstanding characteristics, such as high biocompatibility, biodegradability, antimicrobial property, capacity, biocompatibility, and low toxicity for adsorption, allowing them to be used in pharmaceutical, food, edible film, and biomedical domains [11,12]. Chitosan is a copolymer of D-glucosamine and N-acetyl glucosamine, which can be obtained from the depolymerization or hydrolysis of chitosan [13]. Heating asparagine, fructose/glucose, or chitosans alone at 180 °C for 30 min does not produce acrylamide, indicating the need for a dicarbonyl reactant and Strecker degradation [8,14,15]. Chitosans (50–190 kDa) can increase the browning pigment of the Maillard reaction products (MRPs) and decrease the development of acrylamide in 1% asparagine–fructose and asparagine–glucose food model solutions [14,15]. By increasing the molecular weight, the kinematic viscosity was partially increased [15]. Chitosan amino groups can react with carbonyl groups of aldehydes, ketones, or reducing sugars to form sugar–chitosan conjugates, which have been identified as MRPs [16]. Studying the reduction and development of acrylamide and HMF during food system heating and measuring their effect on various components could assist food industries in designing a more feasible plan for mitigating the neo-formed harmful compound of heat-processed food. Mitigation options for acrylamide in food ingredients, as well as processing foods and control or incorporation of other materials, have been guided by Codex [17]. Following this clue, 0.5% chitosan (50–190 kDa) was mixed with 0.5% asparagine and fructose for 30 min, while evaluating the acrylamide and HMF amounts. In the present and previous studies [18], we managed to assess the impact of 0.5% chitosan on the Maillard reaction in two model solutions, in which asparagine–fructose solution was used with two model systems, aqueous solution and pH 6 acetic acid solution. To elucidate these mechanisms, the potential inhibitory effects of chitosan on the development of HMF and acrylamide on the MRPs generated from asparagine and fructose at different heating periods were recorded. Acrylamide, brown pigments, color, HMF, kinematic viscosity, Maillard intermediate compounds, and pH were compared between the aqueous control solution of fructose/glucose, as well as asparagine and samples with added chitosan in 1% acetic acid solution.

## 2. Materials and Methods

### 2.1. Chemicals and Ingredients

Fructose, chitosan (50–190 kDa with deacetylation degrees greater than 75%), and L-asparagine monohydrate were obtained from Sigma-Aldrich (St. Louis, MI, USA). Acrylamide standard (99.9%) was obtained from J.T. Baker (Phillipsburg, NJ, USA). Oasis MCX (3 mL, 0.06 g) and Oasis HLB (6 mL, 0.2 g) solid-phase extraction cartridges were supplied by Waters (Milford, MA, USA). Chemical reagents employed in this research were of analytical grade.

### 2.2. Preparation of Maillard Reaction Model System

To investigate the effects of chitosan on the formation of acrylamide and HMF, a series of model solutions containing chitosan, fructose, and asparagine were prepared by following the method reported by Chang et al. [19]. Briefly, solutions containing (1) 0.5 g asparagine, (2) 0.5 g fructose, and (3) a combination of 0.5 g asparagine and 0.5 g fructose were dissolved in water. Simultaneously, solutions containing (1) 0.5 g chitosan; (2) a combination of 0.5 g asparagine and 0.5 g fructose; and (3) a combination of 0.5 g asparagine, 0.5 g fructose, and 0.5 g chitosan were dissolved in 1% acetic acid, as well. We adjusted the pH of each solution to 5.8 by adding 1 N sodium hydroxide (NaOH) and then to 6.0 by adding 0.001 N NaOH. Each solution was topped up with distilled water to 100 mL. Thereafter, each solution was placed in a dry-bath incubator (DB200-2, Yisheng Technology Idea Strategy Co. Ltd., New Taipei, Taiwan) maintained at 180 °C for 10, 20, and 30 min and then immediately cooled in tap water prior to analysis. Each sample was prepared in triplicate.

### 2.3. Analyses of Maillard Reaction Products (MRPs), Acrylamide, Hydroxymethylfurfural (HMF), Kinematic Viscosity, Reducing Sugar, Asparagine, and pH of Solutions

Each solution was centrifuged at 4 °C for 15 min at 21,900× *g*, and the supernatant was filtered through a 0.45 μm nylon filter and collected for analyses of MRPs, acrylamide, and HMF. The UV-absorbance and browning intensity of MRPs were measured at OD_294_ and OD_420_, as described by Ajandouz et al. [20], using a UV–Vis spectrophotometer (Synergy HT Multi-detection reader, Biotek Instrument, VT, USA), respectively. A proper dilution was proceeded by using distilled water until the absorbance intensity was less than 1.

The acrylamide level from Maillard reaction was analyzed by HPLC, using the method described by Chang et al. [15]. The HLB/MCX cartridge was preconditioned with 5 and 3 mL of methanol, followed by 5 and 3 mL of deionized, distilled water. The supernatant was passed through a 0.45 μm nylon filter, and 3 mL of the filtrate was then eluted through the preconditioned Oasis HLB/MCX cartridge to absorb acrylamide. The cartridge was first washed with 0.5 mL of deionized, distilled water and the eluate was discarded. Then the sorbent acrylamide was washed with 3.0 mL of deionized, distilled water to clean up the other materials, collected in an amber glass tube, and concentrated under vacuum at 40 °C (RV 10 digital, IKA, Staufen im Breisgau, Germany) for HPLC analysis. The HPLC system (D200) consisted of an L-2130 pump, L-2300 temperature-controlled column oven, L-2200 autosampler, and L-2400 detector (Merck, Hitachi, Kent, UK). The chromatographic HPLC separations were equipped with a COSMOSIL 5C 18-PAQ (5 μm, 4.6 mm × 250 mm) (Nacalai Tesque, Kyoto, Japan), using deionized, distilled water at a flow rate of 0.5 mL/min at 40 °C. The mobile phase was methanol (0.5%), and the filtered concentrated elute was injected by an auto-sampler at 10 °C. Acrylamide quantification was determined from a calibration curve that was built in the range of 0–3125 ppb, using a UV detector at 210 nm.

The supernatant extraction from MRP solution was also measured with the above HPLC system, using the method of Lin et al. [18]. Each MRP solution was vortexed for 60 s. The MRP solution was placed in an ultrasonic water bath for 60 min and separated by using cold centrifugation at 21,900× *g* for 15 min at 4 °C. HMF extraction from MRP solutions (5 μL) was eluted with 5% methanol in water at a flow rate of 0.5 mL/min, at 40 °C, under isocratic conditions. HMF was detected and quantified from the calibration curve built with HMF standard solutions in the range of 0.48–750 ppm, using a UV detector at 284 nm.

The determination of the kinematic viscosity was performed according to the time needed for sample solutions to flow through the capillary viscometers (Cannon-Fenske, No. 50 and No. 75, Cannon Instrument Company, State College, PA, USA). The capillary viscometer was equilibrated in a water bath (Tamson TMV-40, Zoetermeer, The Netherlands) at 30 °C for 10 min. The time required for a solution to drain by gravity through the viscometer was recorded, and this time was converted to a value for kinematic viscosity (cSt), using the extrapolated constant for No. 50 and No. 75 capillary viscometers. To obtain kinematic viscosity, the efflux time was multiplied by the viscometer constant (cSt/s) [21].

Reducing sugars were determined by a dinitrosalicylic acid–reducing sugar assay, following the method of Başkan et al. [22]. Reducing sugar solutions for linearity calibration were evaluated in triplicate by diluting the fructose standard solution, and they contained 0, 375, 750, 1500, 3000, and 6000 μg/mL of fructose standard, using a multi-detection reader (Synergy HT, BIOTEC instrument, Winooski, VT, USA) at 540 nm. The absorbance can be recalculated to the reducing sugar of fructose calibration curve built in the range of 0–6000 μg/mL.

Asparagine in the MRPs solutions was determined by the above HPLC system, as described by Bartolomeo and Maisano [23]. Sample solution (0.5 g) was transferred into a 50 mL centrifuge tube containing 20 mL of 0.1 N HCl. The tube was transferred into an ultrasonic water bath for 10 min and added up with 0.1 N HCl to 25 mL. The solution was filtered through filter paper, and a 20 μL aliquot of the filtrate was transferred into a glass reaction vial. Then 0.4 M borate buffer (100 μL; pH 10.2) was mixed and vortexed. The mixture was added into a glass reaction vial, and 20 μL of o-phthaladehyde (OPA) was mixed and vortexed for 60 s. Then 9-fluoremenylmethyl chloroformate (20 μL; FMOC-Cl) was mixed and vortexed for 30 s. The solution was added up with deionized, distilled water (1280 μL) and the mixture was subjected to derivatization. An amount equivalent to 2.5 μL of the derivatized sample was injected into a Capcell Pak C18 AQ S5 column (5 μm, 4.6 mm × 250 mm) (Shiseido, Tokyo, Japan) at 40 °C. Asparagine was detected at a wavelength of 338 nm of the photodiode array detector for detection. Mobile phase A (40 mM NaH_2_PO_4_) was adjusted to pH 7.8 with NaOH, and mobile phase B consisted of acetonitrile/methanol/water, 45/45/10 *v*/*v*/*v*. The separation was obtained at a flow rate of 2 mL/min, with a gradient program that allowed for 0.5 min at 0% B flowed by a 13.0 min step that increased eluent B to 46%. Then washing at 100% B and equilibration at 0% B was performed in a total analysis time of 20 min.

The pH values of the sample solution were measured by using a pH meter (pH 510, Eutech Instruments Pte Ltd., Singapore) calibrated with a buffer solution of pH 4.0 and 7.0 (AppliChem GmbH, Darmstadt, Germany).

### 2.4. Chromaticity Testing

A spectrophotometer (TC-1800 MK II, Tokyo, Japan) was used to measure the color of the solution sample, using International Commission on Illumination (CIE), based on L* (lightness/darkness), a* (redness/greenness), and b* (yellowness/blueness) values. The spectrophotometer was standardized against a white tile and black cup before the test. Each solution was measured 3 times and in triplicate. Total color difference (ΔE) was calculated by using the following equation:ΔE = [(△L*)^2^ + (△a*)^2^ + (△b*)^2^]^1/2^
where △L* = L*_sample_ − L*_control_, △a* = a*_sample_ − a* _control_, and △b* = b* _sample_ − b* _control_.

CIE L*_control_ a*_control_ b*_control_ was the value for the fructose–asparagine MRP solutions.

### 2.5. Statistical Analysis

Data were analyzed by using the software Statistic Package for Social Science (SPSS 2000) statistics program for Window Version 12 (SPSS Inc., Chicago, IL, USA). All reactions were performed in triplicate. For statistical purposes, values below the limit of detection were replaced by zero. Statistical differences were considered with *p* < 0.05 by using analysis of variance (ANOVA) and Duncan’s test. The results were expressed as means ± standard deviation. Pearson’s correlation coefficients were used to evaluate the linear correlations of different functional properties at *p* < 0.05 and *p* < 0.01 significance levels.

## 3. Results

### 3.1. MRPs of Solutions

Figure 1 and Figure 2 show the formation of the Maillard reaction products at OD_2__94_ and OD_420_, respectively, in the water and acetic acid solutions. OD_294_ indicates the formation of the Maillard reaction intermediates, whereas OD_420_ indicates the formation of the brown intensity of the Maillard reaction products [24,25]. In this study, the intensity of OD_294_ was higher than that of OD_420_ in a heated solution. Fructose and asparagine dissolved in 1% acetic acid had higher absorbance intensities than solutions dissolved in distilled water at either OD_294_ or OD_420_ after the heating process. As shown in Figure 1A, the heating process caused a significant change in the absorbance of asparagine–fructose after heating for 20 min or more, but not in asparagine and fructose. In the acetic acid solution, the heating process of fructose–asparagine or fructose–asparagine–chitosan solution for 20 min or more caused a significant increase in absorbance value (Figure 1B). Fructose–asparagine and fructose–asparagine–chitosan had the highest absorbance at OD_294_ among all groups after heating for 30 min, which has a value of 33.7 ± 1.1 and 33.7 ± 4.77, respectively.

### 3.2. Effect of Chitosan Addition on the Formation of Acrylamide and HMF

Figure 3 presents the amounts of acrylamide development for various groups upon heating in water and acetic acid. In this study, we did not observe the presence of acrylamide in either asparagine and fructose dissolved in water or chitosan dissolved in acetic acid in any heating period (Figure 3). Conversely, acrylamide was found in asparagine–fructose dissolved in water after heating for 20 min or more (Figure 3A). In the acetic acid solution, this compound was detected in asparagine–fructose after being heated for 10 min or more and in asparagine–fructose–chitosan after being heated for 20 min or more (Figure 3B). The heating of asparagine–fructose solution for 30 min generated 1859 and 9401 ppb in water and acetic acid, respectively. While the heating process significantly increased the acrylamide content of asparagine–fructose dissolved in acetic acid after heating the solution for 20 min or more (Figure 3B), it did not occur that much for asparagine–fructose dissolved in water even after heating for 30 min (Figure 3A). Furthermore, we observed that the acrylamide production in asparagine–fructose–chitosan dissolved in acetic acid was relatively lower than that of asparagine–fructose dissolved in the same solution, and the difference was significant (Figure 3B), thus indicating that chitosan is capable of inhibiting the production of acrylamide from heating reducing-sugar.

In contrast to the gradual accumulation (ppb) of acrylamide, high amounts of HMF were detected in the heated model systems of fructose/glucose and asparagine–fructose/glucose after 10 min of heating, except for 0.5% asparagine (Figure 4). It should be noted that the content of acrylamide was measured at the ppb level, whereas HMF was at the ppm level. The HMF concentrations were higher in model systems containing 0.5% asparagine–fructose/glucose and fructose/glucose dissolved in distilled water after heating than those in a solution containing 0.5% asparagine–fructose/glucose and asparagine–fructose/glucose–chitosan dissolved in 1% acetic acid. HMF (314 ppb) was detected in 0.5% chitosan solution after heating for 30 min (Figure 4B). The content of HMF was higher than the content of acrylamide in MRPs.

Gökmen and Senyuva [26] reported that HMF can react with asparagine and generate acrylamide, as was also confirmed in our previous study [18]. In this study, the addition of chitosan did not enhance the formation of HMF (Figure 4B). Our previous study showed that the HMF concentrations in 0.5% HMF or HMF–asparagine solutions did not change considerably after heating (*p* > 0.05) [18]. As HMF can react with asparagine to form MRPs and acrylamide, the production of HMF is probably generated via the caramelization of fructose/glucose (91.56 ppm (Figure 4A) and 34.51 ppm [18]) after heating for 30 min, as compared to HMF generated by the formation of a dicarbonyl intermediate and 3-deoxyglucosone from the Maillard reaction (12.28 ppm (Figure 4A) and 5.13 ppm [18]).

### 3.3. Kinematic Viscosity, Reducing Sugar, Asparagine, and pH of Solutions

Figure 5 shows the changes in the kinematic viscosity of the solutions tested in this study. The kinematic viscosity of the solutions without chitosan remained constant even after 30 min of heating (Figure 5). However, the kinematic viscosity of 0.5% chitosan or asparagine–fructose–chitosan decreased significantly during 10–30 min of heating to a range of 1.24–1.84 cSt, thus indicating that chitosan was hydrolyzed. Asparagine–fructose, asparagine, and fructose dissolved in 1% acetic acid or water showed kinematic viscosity in the range of 0.81–0.83 cSt. Moreover, the viscosity of solutions containing 0.5% chitosan and asparagine–fructose–chitosan is in the range of 5.49–5.91 cSt.

Figure 6 shows the reducing sugar content (RSC) in various solutions tested in this study. The amount of RSC in all types of solutions dissolved in water or acetic acid did not change significantly, even after heating for 30 min. Fructose (0.5%) and asparagine–fructose solutions heated for 30 min contain 4680 and 4262 μL/mL reducing sugar, respectively (Figure 6A). In the acetic acid solution, the chitosan addition failed to change the amount of RSC in asparagine–fructose–chitosan solution (Figure 6B). Nevertheless, heating asparagine–fructose or asparagine–fructose–chitosan for 10 min significantly produced a higher amount of RSC than those detected in unheated chitosan and chitosan solution heated for 10 min. A slight increase in the RSC value of asparagine–fructose from 5061 to 5445 g/mL was observed in this study after heating for 10 min and a decrease to 2191 g/mL after 30 min of heating (Figure 6B). The level of RSC in chitosan–fructose–asparagine was increased from 5543 to 6158 μg/mL after heating for 10 min, but then was decreased significantly to 3756 μg/mL after 30 min of heating.

Figure 7 presents the heating effect on the asparagine content in solutions tested in this study. A high level of asparagine content was found in all unheated groups, ranging from 4261 to 5135 μg/mL. None of the heating treatments can significantly reduce the asparagine level in either asparagine–fructose or asparagine alone dissolved in water (Figure 7A), indicating the absence of asparagine deamidation to aspartic acid. In acetic acid solution, heating the solution for 30 min can significantly reduce the asparagine content of the asparagine–fructose and asparagine–fructose–chitosan solution, indicating that the amount of asparagine did not change after the addition of chitosan (Figure 7B). The level of asparagine content was reduced from 4687 to 2322 μg/mL and from 4495 to 2582 μg/mL in asparagine–fructose and asparagine–glucose–chitosan dissolved in acetic acid, respectively, after heating for 30 min (Figure 7B).

A larger reduction in pH was observed in fructose dissolved in water after heating for 30 min (Figure 8A). In acetic acid solution, the pH of 0.5% chitosan remained constant after heating for 30 min (Figure 8B). However, a significant reduction of pH was observed in asparagine–fructose and asparagine–fructose–chitosan dissolved in acetic acid after being heated for 30 min. In addition, this study did not observe the difference in the pH value between asparagine–fructose and asparagine–fructose–chitosan dissolved in acetic acid in each heating-time point, indicating that the chitosan addition did not affect the pH change (Figure 8B).

### 3.4. The Colorimetric Analysis of MRPs

The colorimetric analysis L*a*b* values of different combinations of fructose, asparagine, and chitosan-heated solutions, are shown in Table 1 and Table 2. The yellowness (b* value) of fructose–asparagine and fructose–asparagine–chitosan increased and the lightness (L* value) decreased as the heating time increased (Table 1 and Table 2 and Appendix A). However, the redness (a* value) in Table 1 and Table 2 does not follow the trend. The L* value is negatively correlated with acrylamide content (r = −0.923) and MRPs’ brown pigment (Appendix A; *p* < 0.01; r = −0.969). However, the ΔE values are positively correlated with those of a* and b* values (Appendix A; *p* < 0.01).

There was a high positive correlation between acrylamide and HMF (Appendix A; *p* < 0.01) when chitosan was added to the fructose/glucose model system. The heating time is positively related to the acrylamide contents (Appendix A), but not to HMF. The acrylamide formation of MRP solution is positively correlated to the heating time (Appendix A; r = 0.434; and r = 0.463, *p* < 0.05). The HMF content increased upon the formation of Maillard intermediate compounds when the ingredients were dissolved in 1% acetic acid (r = 0.960) and water (r = 0.999) (Appendix A). The HMF concentration increased with OD_420_ when the ingredients were dissolved in 1% acetic acid (r = 0.899 and 0.885) (Appendix A) in the fructose/glucose model systems.

## 4. Discussion

### 4.1. MRPs of Solutions

The intermediate compounds of MRPs increase in both models when heated for 30 min in a reducing sugar-containing system dissolved in 1% acetic acid, especially for dissolving chitosan, compared to the model of the reducing sugar-containing system dissolved in distilled water. This implies that the Maillard reaction occurs easily in the reducing sugar dissolved in 1% acetic acid. Mengibar et al. [27] demonstrated that conjugates generated from enzymatically depolymerized chitooligosaccharide increase in OD_294_ and OD_420_ (*p* < 0.05); the results of heated 0.5% chitosan (50–190 kDa) only significantly increase (*p* < 0.05) in OD_294_ and OD_420_ of the glucose–asparagine model system after heating for 30 min. Li et al. [28] also showed that chitosan (190 kDa) added to maltose increases the formation of MRPs. The addition of heated 1% chitosan (50–190 kDa) enhanced the intensity of OD_294_ and OD_420_ of asparagine–fructose solution (AFC) [15]. In this study, the addition of 0.5% chitosan (50–190 kDa) did not increase the intensity of OD_294_ and OD_420_ in 0.5% fructose with 0.5% asparagine solution. However, a previous study also showed that the addition of chitosan also increased the production of the intermediate compound and brown MRP pigment in 0.5% glucose with 0.5% asparagine solution [18]. The absorbance values of OD_294_ and OD_420_ in fructose-containing solutions (Figure 1 and Figure 2) were higher than those of glucose-containing solutions reported in our previous study [18]. In addition, the ultraviolet-absorbing intermediate formation could not reach the maximum after heating for 30 min in this study (Figure 1). In our previous study, we found that the intensities of OD_294_ and OD_420_ gradually increased with the heating time in solutions of fructose/glucose, asparagine–fructose/glucose, and asparagine–fructose/glucose–chitosan [18]. Another study reported that OD_294_ can increase at first and then decrease after prolonged heating [29].

### 4.2. Effect of Chitosan Addition on the Formation of Acrylamide and HMF

The solutions containing fructose and asparagine with or without chitosan were heated for 30 min to obtain the kinetic behavior of acrylamide and HMF. The kinetic approaches involving HMF and chitosan can help create a better understanding of the formation of MRPs. We have confirmed that the addition of chitosan reduced the formation of acrylamide in the reducing sugar fructose (keto) system (Figure 3B) but increased the formation of acrylamide after heating for 30 min in the glucose (Aldo) system [18]. The 1% low-molecular chitosan and 1% chitooligosaccharide were observed to reduce the production of acrylamide in 1% asparagine and glucose solutions, respectively [14,19]. This could be due to the extremely high amount of acrylamide (>80,000 ppb) generated in those studies compared to 2370 ppb acrylamide formed in this study. However, more acrylamide (>8000 ppb) was formed in the system that was first dissolved in 1% acetic acid (Figure 3B). This is the first time that asparagine–glucose–chitosan has been observed to promote the generation of acrylamide after heating for 30 min at 180 °C [18]. The group with the chitosan addition after heating for 30 min resulted in a lower amount of acrylamide and HMF content, making it decrease acrylamide generation in the solution with fructose as reducing the sugar by adding 0.5% chitosan, but not glucose system.

The free amines of hydrolyzed 0.5% chitosan and chitooligosaccharide in the solution can compete with free asparagine. In contrast to the effect of 0.5% mixture asparagine–glucose–chitosan solution, the effect of chitosan in mitigating the formation of acrylamide was statistically significant (*p* < 0.05) in 0.5% asparagine–fructose–chitosan mixture solution in this study during the first 20 min of heating (Figure 3B). However, low-molecular-weight chitosans or chitooligosaccharides increased the production of acrylamide [18]. This was contrary to what was observed in previous studies [14,19]. However, all MRPs’ content increased after heating for 30 min (Figure 1 and Figure 2). This can support the 0.5% chitosan degraded to lower-molecular-weight chitosan, chitooligosaccharide, or glucosamine in the solution after heating for 30 min to react with asparagine and generate less acrylamide (Figure 3B) in the asparagine–fructose model system.

Gökmen and Senyuva [26] reported that the addition of sodium ions before thermal processing enhanced the generation of HMF and acrylamide in a glucose–asparagine solution. This is similar to the 0.5% fructose and asparagine dissolved in 1% acetic acid and then adjusted with sodium hydroxide to modify to pH 6 in this study. The activation energy of acrylamide formation was 83.94 and 138.78 kJ/mol for the model systems of asparagine–glucose and asparagine–HMF, respectively [30]. Gökmen et al. [30] claimed that the acrylamide formation obeys the Arrhenius law, with high-correlation coefficients from 90 to 180 °C. HMF is an intermediary compound of MRP, and it can be subjected to further reactions [31]. Qi et al. [31] reported that the HMF content in glucose to asparagine solutions increased followed by a decreasing trend, which was different from the gradual increase of acrylamide. Therefore, the reaction of HMF with asparagine is not the major pathway for acrylamide development [18], and it was confirmed in this present study.

All test model systems with 0.5% fructose/glucose, asparagine, and chitosan formed lower acrylamide, HMF, and MRPs content, respectively, after heating for 30 min than the results with 1% mixture [14,18,19]. Therefore, acrylamide is formed mainly from the reaction of glucose with asparagine [26,32]; however, glucosamine degraded from chitosan after heating for 30 min may mitigate acrylamide formation. Heating asparagine powder alone could generate acrylamide via thermal degradation [33]. However, acrylamide was not detected in heated 0.5% chitosan or asparagine solutions after 30 min of heating in this study (Figure 3). Charoenprasent et al. [32] found that glucosamine was added to California-style black ripe olives before sterilization. Charoenprasert et al. [32] claimed that N-acetyl glucosamine and glucosamine were the major amino sugars detected in olives. These two compounds contain carbon and nitrogen atoms and have reactive sites, such as amine, carbonyl, and hydroxyl, which can enable the development of acrylamide in California-style black ripe olives by intrachemical and interchemical reactions.

Glucose can react with asparagine to form a Schiff base, which can rearrange to give an Amadori product. The Amadori product dehydrates, and the fragments form highly reactive deoxyglucose, hydroxycarbonyl compounds, or dicarbonyl compounds. The dicarbonyl compounds react with asparagine via Strecker degradation, leading to the production of acrylamide. Acrylamide is generated when glucose reacts with asparagine to generate a Schiff base that is decarboxylated to generate acrylamide, without fragmentation of sugar and rearrangement of Amadori products.

Gökmen et al. [30] proposed that an equimolar asparagine–HMF low-moisture system produced acrylamide more easily than the asparagine–glucose model system when heated at 180 °C. The HMF and other carbonyls from sugar dehydration cannot generate acrylamide in the thermally processed food model in our previous study [18].

There was a negative correlation between asparagine concentration and HMF formation rates, thus showing another chemical pathway for HMF formation [34]. The pH of the asparagine–fructose/glucose solution [18] was 6.55 after heating for 30 min, which was higher than that (pH 4.41) of 0.5% fructose/glucose alone after 30 min of heating (Figure 8A). The heated 0.5% fructose/glucose after 30 min demonstrated that more HMF was formed at lower pH values than the solutions of 0.5% asparagine and fructose/glucose [18], and this may be due to HMF being prone to generate at acidic pH.

More HMF was formed at low pH conditions from fructose and glutamic acid in the model system [35]. HMF has been confirmed to be mainly derived from the caramelization of reducing sugars [2], as was also agreed on in the results of Lin et al. [18]. Because HMF is reactive and has a high concentration (0.5% glucose and asparagine), sugars could be transferred to HMF at a lower rate than its elimination when chitosan was included. Chitosan can be hydrolyzed into lower-molecular-weight chitosan and lead to other compounds, including HMF (Figure 4B). Locas and Yaylayan [36] proposed that reducing sugar, carbonyl groups of ketones, or aldehydes can react with amino groups of chitosan to form chitosan–sugar conjugates and HMF of MRPs. The mechanism is different from the formation of acrylamide in MRPs produced from asparagine and glucose. Therefore, the carbonyl group of reducing sugars, aldehyde, and glucosamines or ketones may react with the amino group of glucosamine to form HMF and acrylamide. HMF is an intermediate, which can also react with amines of itself. The results in Figure 4B confirmed that the hydrolyzed chitosan reacts to mitigate the formation of HMF, especially in the fructose–asparagine model system after 30 min of heating. Chitosan may hydrolyze to release free glucosamine as a reactive intermediate. Locas and Yaylayan [36] reported that fructose/glucose in the presence of water can be converted into glucose, and it can be protonated to a fructofuranosyl cation and reacts with asparagine to form fructofuranosyl amine, which can rearrange into a Heyns product before acrylamide forming. Fructofuranosyl cation may have reacted with chitosan, chitooligosaccharides, or glucosamine in this study. A high amount of HMF was detected in a heated model system of asparagine and fructose dissolved in water. The formation of HMF at a high level was observed in the 0.5% fructose model system (Figure 4A), which was a different reaction pathway from Maillard reactions. The carbonyl group of the reducing sugar fructose can react with the amino group of chitosan and chitooligosaccharide to mitigate the formation of chitosan–fructose or chitooligosaccharide–fructose conjugates, and HMF of MRPs, as well as the conjugates that would prevent the formation of HMF during the heating process. Therefore, the heating time for the formation of HMF and acrylamide in fructose/glucose model solutions should be less than 10 min. When chitosan is added to mitigate acrylamide formation, the heating time should not exceed 20 min.

### 4.3. Kinematic Viscosity, Reducing Sugar, Asparagine, and pH of Solutions

Low-concentration chitosan exhibits Newtonian behavior. When the concentration of chitosan exceeds 0.5%, it becomes a non-Newtonian fluid [37]. The kinematic viscosity of asparagine–fructose/glucose–chitosan and chitosan decreased significantly after heating for 10 min at 180 °C (Figure 5). Therefore, we assumed that chitosan was potentially vulnerable to glucosamine, chitooligosaccharide, or low-molecular-weight chitosan after 30 min of heating. N-acetyl glucosamine and D-glucosamine can be formed by depolymerization or hydrolysis of chitosan [13]. The viscosity of chitosan (1%) in 1% acetic acid reduced to 91% of the original viscosity after heating for 15 min in an autoclave [38]. Chitosan was hydrolyzed to glucosamine in acetic acid and acetyl glucosamine when the hydrolysis was incomplete. Additionally, it was hydrolyzed into chitooligosaccharide in weak acid at low temperatures [39].

When comparing asparagine–glucose and asparagine–fructose dissolved in deionized water samples with those dissolved in 1% acetic acid, the decrease in reducing sugar content (RSC) after heating for 30 min was significant (Figure 6), which could be due to sodium chloride formed during the neutralized process. Perhaps, glucose was degraded on the Maillard reaction under pH control during heating [40]. A low level of reducing sugar (125 μg/mL) was detected in 0.5% chitosan solution, and it increased to 225 μg/mL (*p* < 0.05) after 30 min of heating (Figure 6B). Doolittle et al. [41] proposed that this may be because chitosan is hydrolyzed into glucosamine and chitooligosaccharide, and their acetyl groups can react with dinitrosalicylic acid to increase the absorbance during the reducing assay. Yan and Evenocheck [39] also reported that the RSC of heated 0.5% chitosan solution increased could be due to chitosan hydrolysis. Yan and Evenocheck [39] also proposed chitosan hydrolysis at 90 °C in a weak acid, with oligosaccharides as the main hydrolysate products. Chitosan may hydrolyze to glucosamine in acetic acid. Yan and Evenocheck [39] reported that acetyl glucosamine was detected in an incomplete hydrolysate product. Reducing sugar was also found in HMF solution and HMF–asparagine solution [18] but less than that of glucose [18] and fructose (Figure 6A). Deshavath et al. [42] reported that HMF and furfural in solutions increase absorbance in 3,5-dinitrosalicylic acid (DNS) assay for reducing sugars. The content of reducing sugars of HMF and HMF–asparagine by DNS assay does not change significantly after heating for 30 min [18].

The asparagine in the asparagine-alone groups in deionized water heated for 30 min decreased less than it did in all other heated mixtures (Figure 7A), indicating that the formation of MRPs deprived more asparagine with reducing sugar than heating asparagine alone (Figure 7A). Ajandouz et al. [20] also reported that fructose loss was greater in the presence of lysine after heating for 5 min than in heated fructose alone.

Kavousi et al. [35] reported that HMF hydrolyzed to levulinic and formic acids and released protons, thereby decreasing the pH of heated solutions. De Vleeschouwer et al. [43] confirmed that the deamidation of asparagine to aspartic acid causes a pH increase when the solution is heated at 180 °C. The carbonyl sources in aldehydes could interact with the ketones of reducing sugars and decrease the solution’s pH. The pH of fructose/glucose and asparagine solutions dissolved in 1% acetic acid and then adjusted back to pH 6 or in distilled water differs after 30 min of heating, but not the pH of 0.5% chitosan solution (Figure 8). Martins and Van Boekel [44] proposed that acetic acid, formic acid, glyoxal, and pyruvaldehyde formed during the Maillard reaction, but they decrease the pH of a comparable trial solution.

The pH of hydrolyzed chitosans is unaffected (Figure 8B), but the higher amounts of acrylamide in the glucose model system and MRPs, as well as lower viscosity of solutions generated in both chitosan trials, could be due to glucosamine or oligochitosan, which can easily participate in the Maillard reaction compared to chitosans to decrease the pH of the asparagine–fructose/glucose–chitosan solution.

### 4.4. Chromaticity Testing of MRPs

The increase in the b* value and ΔE of 0.5% chitosan solutions suggested the occurrence of the Maillard reaction (Table 1 and Table 2). This browning activity correlated significantly with the effect of hydrolyzed chitosan on acrylamide formation and MRPs in this study. It was also consistent with an increase in OD_294_ and OD_420_ absorbance (*p* > 0.05). Therefore, we proposed the addition of chitosan hydrolyzed to chitooligosaccharide, acetylglucosamine, and glucosamine to the Maillard reaction. These results were also observed in previous studies [14,15,18,19].

## 5. Conclusions

Asparagine, chitosan, and fructose (0.5%) were used to elucidate the possible reactions between these molecules at various heating intervals and ingredient-dissolved settings in the ketose and aldose sugars designed model system. This study evaluated the acrylamide content and functional properties of MRPs generated in the heated solution. Chitosan contains amino groups that can react with the carbonyl groups of fructose to form MRPs and mitigate 48% acrylamide formation at 30 min heating stage in the fructose–asparagine model system. HMF was generated during the first 20 min of heating in solution and chitosan heated alone for 20 min. The kinematic viscosity of the chitosan-containing model system was decreased significantly after heating for 10 min, and the hydrolyzed chitosan caused the generation of acrylamide and a significant reduction of HMF after heating for 20 min in the glucose model system, but not in the fructose model system. The amino groups of hydrolyzed chitosan can compete with asparagine and also react with the carbonyl groups of fructose to generate more chitosan–MRPs, but mitigating the formation of acrylamide. However, the formation of carcinogenic acrylamide is greater than 1000 ppb after heating for 20 min in the fructose–asparagine solution system. Moreover, the heating process significantly increased the acrylamide content of asparagine–fructose dissolved in acetic acid system when compared to acrylamide generated in the heated aqueous asparagine–fructose system. The formation of HMF was greater than 50 ppm in fructose solution after heating for 30 min, which should be avoided for food processing. The fructose-containing heated solution has more potential for forming MRPs, such as acrylamide and HMF, which could be mitigated by the addition of chitosan, but not by heating for more than 20 min, as this creates safety issues.

## Figures and Tables

**Figure 1 polymers-14-01565-f001:**
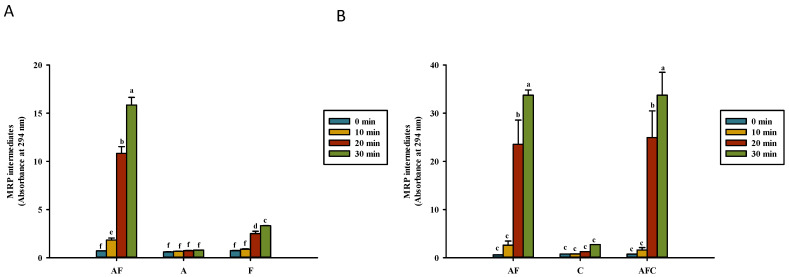
Maillard reaction product (MRP) intermediate development (OD_294_) for various groups upon heating in (**A**) water and (**B**) acetic acid. AF, asparagine and fructose; A, asparagine; F, fructose; C, chitosan; AFC, asparagine, fructose and chitosan. Data represent the means of three replicates, and error bars indicate standard deviation. Bars followed by a common letter are not significantly different according to the Dunn’s test at the 5% level of significance.

**Figure 2 polymers-14-01565-f002:**
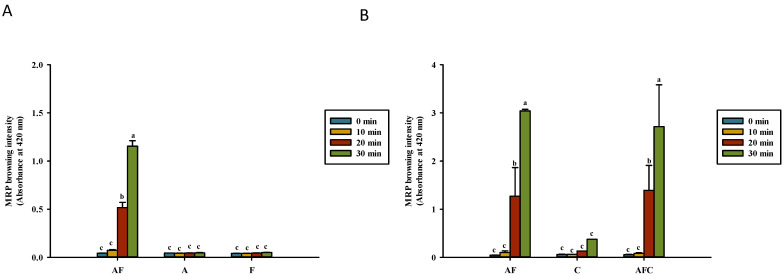
Brown color development (OD_420_) of Maillard reaction product (MRP) for various groups upon heating in (**A**) water and (**B**) acetic acid. AF, asparagine and fructose; A, asparagine; F, fructose; C, chitosan; AFC, asparagine, fructose, and chitosan. Data represent the means of three replicates, and error bars indicate standard deviation. Bars followed by a common letter are not significantly different according to the Dunn’s test at the 5% level of significance.

**Figure 3 polymers-14-01565-f003:**
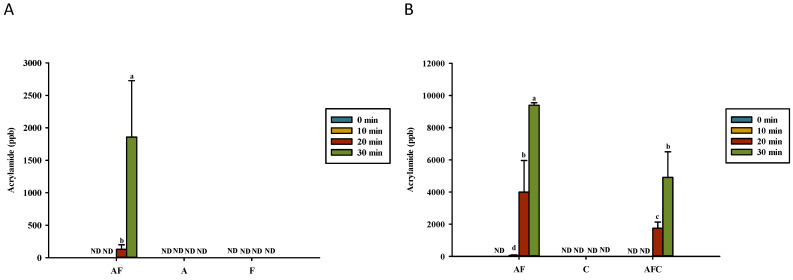
Amounts of acrylamide development for various groups upon heating in (**A**) water and (**B**) acetic acid. AF, asparagine and fructose; A, asparagine; F, fructose; C, chitosan; AFC, asparagine, fructose and chitosan. Data represent the means of three replicates, and error bars indicate standard deviation. Bars followed by a common letter are not significantly different according to the Dunn’s test at the 5% level of significance (ND, not detected).

**Figure 4 polymers-14-01565-f004:**
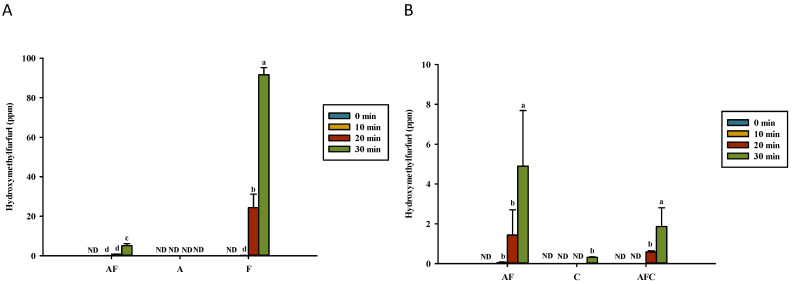
Hydroxymethylfurfural formation for various groups upon heating in (**A**) water and (**B**) acetic acid. AF, asparagine and fructose; A, asparagine; F, fructose; C, chitosan; AFC, asparagine, fructose and chitosan. Data represent the means of three replicates, and error bars indicate standard deviation. Bars followed by a common letter are not significantly different according to the Dunn’s test at the 5% level of significance (ND, not detected).

**Figure 5 polymers-14-01565-f005:**
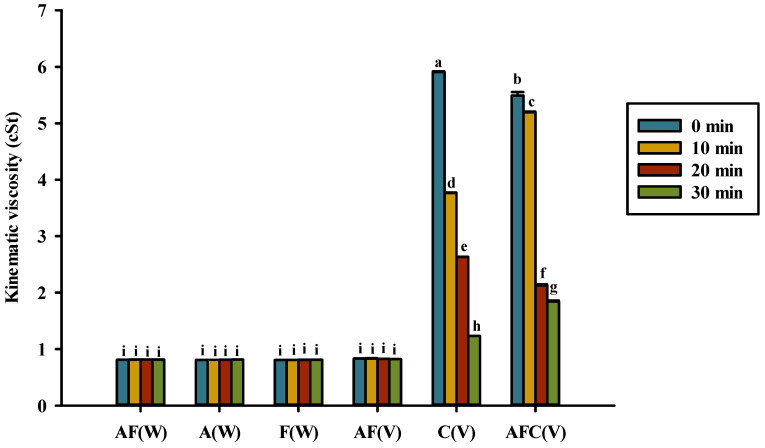
Kinematic viscosity for various groups upon heating in water or acetic acid. AF(W), asparagine and fructose in water; A(W), asparagine in water; F(W), fructose in water; AF(V), asparagine and fructose in acetic acid; C(V), chitosan in acetic acid; AFC(V), asparagine, fructose, and chitosan in acetic acid. Data represent the means of three replicates, and error bars indicate standard deviation. Bars followed by a common letter are not significantly different by the Dunn’s test at the 5% level of significance.

**Figure 6 polymers-14-01565-f006:**
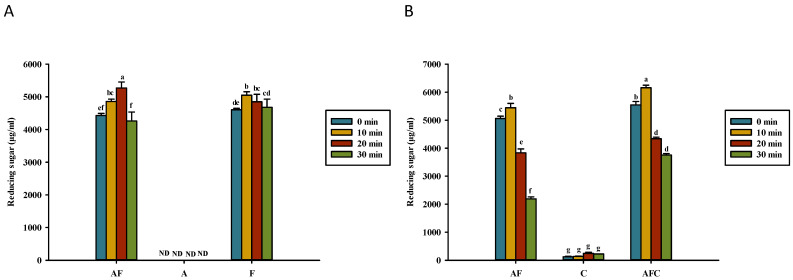
Reducing sugar concentration for various groups upon heating in (**A**) water and (**B**) acetic acid. AF, asparagine and fructose; A, asparagine; F, fructose; C, chitosan; AFC, asparagine, fructose and chitosan. Data represent the means of three replicates, and error bars indicate standard deviation. Bars followed by a common letter are not significantly different according to the Dunn’s test at the 5% level of significance (ND, not detected).

**Figure 7 polymers-14-01565-f007:**
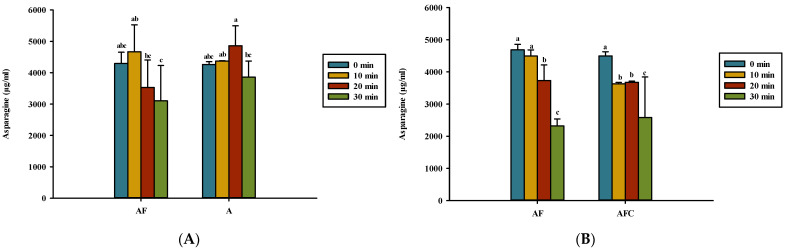
Asparagine content for various groups upon heating in (**A**) water and (**B**) acetic acid. AF, asparagine and fructose; A, asparagine; AFC, asparagine, fructose, and chitosan. Data represent the means of three replicates, and error bars indicate standard deviation. Bars followed by a common letter are not significantly different according to the Dunn’s test at the 5% level of significance.

**Figure 8 polymers-14-01565-f008:**
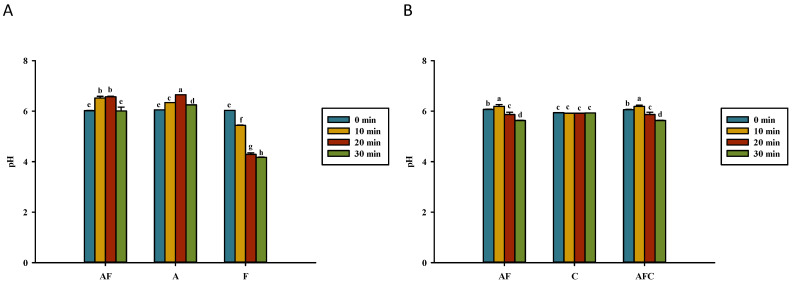
Comparison of pH values for various groups upon heating in (**A**) water and (**B**) acetic acid. AF, asparagine and fructose; A, asparagine; F, fructose; C, chitosan; AFC, asparagine, fructose, and chitosan. Data represent the means of three replicates, and error bars indicate standard deviation. Bars followed by a common letter are not significantly different according to the Dunn’s test at the 5% level of significance.

**Table 1 polymers-14-01565-t001:** Chromatic parameters of solutions containing a combination of asparagine and fructose during 0–30 min of heating in water.

	L*	a*	b*	ΔE
AFW 0 min	100.13 ± 0.00 ^a^	0.42 ± 0.01 ^a^	−0.20 ± 0.04 ^c^	0.48 ± 0.02 ^c^
AFW 10 min	100.13 ± 0.01 ^a^	0.08 ± 0.12 ^cd^	0.41 ± 0.20 ^c^	0.47 ± 0.15 ^c^
AFW 20 min	100.08 ± 0.01 ^bc^	−2.07 ± 0.15 ^e^	4.26 ± 0.25 ^b^	4.73 ± 0.28 ^b^
AFW 30 min	98.97 ± 0.08 ^e^	−17.10 ± 0.25 ^f^	47.01 ± 1.66 ^a^	50.04 ± 1.65 ^a^
AW 0 min	100.10 ± 0.01 ^ab^	0.31 ± 0.01 ^ab^	−0.11 ± 0.01 ^c^	0.35 ± 0.00 ^c^
AW 10 min	100.11 ± 0.01 ^ab^	0.38 ± 0.03 ^ab^	−0.19 ± 0.04 ^c^	0.44 ± 0.05 ^c^
AW 20 min	100.11 ± 0.01 ^ab^	0.41 ± 0.03 ^a^	−0.19 ± 0.05 ^c^	0.46 ± 0.05 ^c^
AW 30 min	100.12 ± 0.00 ^ab^	0.34 ± 0.02 ^ab^	−0.08 ± 0.06 ^c^	0.37 ± 0.03 ^c^
FW 0 min	100.00 ± 0.01 ^d^	0.12 ± 0.03 ^cd^	−0.07 ± 0.06 ^c^	0.14 ± 0.05 ^c^
FW 10 min	100.05 ± 0.02 ^c^	0.32 ± 0.11 ^ab^	−0.24 ± 0.15 ^c^	0.40 ± 0.18 ^c^
FW 20 min	100.06 ± 0.01 ^c^	0.23 ± 0.02 ^bc^	−0.04 ± 0.03 ^c^	0.24 ± 0.02 ^c^
FW 30 min	100.08 ± 0.01 ^bc^	0.06 ± 0.07 ^d^	0.32 ± 0.08 ^c^	0.34 ± 0.07 ^c^

Values are mean ± standard deviation of three replicates. Values followed by a common letter in the same column are not significantly different by the Dunn’s test at the 5% level of significance. AFW, asparagine and fructose; AW, asparagine; FW, fructose.

**Table 2 polymers-14-01565-t002:** Chromatic parameters of solutions containing the combination of asparagine, fructose, and chitosan dissolved in 1% acetic acid and titrating back to pH 6.

	L*	a*	b*	ΔE
AF 0 min	100.14 ± 0.01 ^a^	0.42 ± 0.02 ^c^	10.78 ± 0.08 ^fg^	10.79 ± 0.08 ^fg^
AF 10 min	100.05 ± 0.01 ^b^	−3.77 ± 0.02 ^e^	7.57 ± 0.03 ^gh^	8.46 ± 0.03 ^gh^
AF 20 min	95.72 ± 0.01 ^h^	−14.44 ± 0.02 ^k^	68.61 ± 13.31 ^d^	70.30 ± 13.00 ^d^
AF 30 min	62.54 ± 0.03 ^j^	20.82 ± 0.03 ^a^	99.35 ± 0.07 ^a^	108.20 ± 0.06 ^a^
C 0 min	100.08 ± 0.01 ^b^	−1.19 ± 0.04 ^d^	2.52 ± 0.07 ^hi^	2.79 ± 0.08 ^h^
C 10 min	99.81 ± 0.00 ^d^	−5.82 ± 0.02 ^g^	9.22 ± 0.06 ^fg^	10.90 ± 0.06 ^fg^
C 20 min	99.79 ± 0.12 ^d^	−8.03 ± 0.05 ^i^	14.27 ± 0.09 ^f^	16.38 ± 0.10 ^f^
C 30 min	99.03 ± 0.01 ^f^	−12.47 ± 0.03 ^j^	25.71 ± 0.05 ^e^	28.59 ± 0.03 ^e^
AFC 0 min	99.66 ± 0.04 ^e^	−7.16 ± 0.06 ^h^	−0.16 ± 0.03 ^i^	7.17 ± 0.06 ^gh^
AFC 10 min	99.98 ± 0.01 ^c^	−4.18 ± 0.02 ^f^	7.52 ± 0.03 ^gh^	8.60 ± 0.03 ^gh^
AFC 20 min	95.86 ± 0.06 ^g^	−14.36 ± 0.47 ^k^	83.89 ± 0.02 ^c^	85.21 ± 0.06 ^cb^
AFC 30 min	75.21 ± 0.04 ^i^	14.37 ± 0.03 ^b^	91.86 ± 0.09 ^b^	96.23 ± 0.07 ^cb^

Values are mean ± standard deviation of three replicates. Values followed by a common letter in the same column are not significantly different by the Dunn’s test at the 5% level of significance. AF, asparagine and fructose; C, chitosan; AFC, asparagine, fructose, and chitosan.

## Data Availability

The data presented in this study are available on request from the corresponding author.

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
