# Peer review of "Effect of Chitosan Incorporation on the Development of Acrylamide during Maillard Reaction in Fructose–Asparagine Model Solution and the Functional Characteristics of the Resultants"

_polymers, 2022, doi:10.3390/polym14081565_

Round 1
Reviewer 1 Report
The resubmission of paper titled (Effect of chitosan incorporation on the development of acrylamide during Maillard reaction in fructose–asparagine model solution and the functional characteristics of the resultants) was partly improved. The significance symbols on the figures are completely wrong & unreadable or comprehendable.
I believe the authors must consult a statistician to advice them how to present the statistical significance on a graph. Please submit a revised form & confirm in which you have consulted a statistician who helped you in drawing the figures and adding the significance sympols.
Author Response
Dear Reviewer #1
The authors are extremely grateful to anonymous referee involved for providing his/her excellent comments and valuable advice in this paper. We have revised the paper based on the referee’s comments. We have pleasure in requesting the referee to review this paper. Thank you. Your prompt attention to this paper will be much appreciated.
Point 1: The resubmission of paper titled (Effect of chitosan incorporation on the development of acrylamide during Maillard reaction in fructose–asparagine model solution and the functional characteristics of the resultants) was partly improved. The significance symbols on the figures are completely wrong & unreadable or comprehendable.
I believe the authors must consult a statistician to advice them how to present the statistical significance on a graph. Please submit a revised form & confirm in which you have consulted a statistician who helped you in drawing the figures and adding the significance sympols.
Response 1: We are sorry for the mistakes on the significance symbols of figures. We have consulted statisticians Dr. Ndraha and Professor Hsiao and asked them help us redraw the significance symbols on the figures of this work compared at all test solutions upon different heating duration and add “Bars followed by a common letter are not significantly different by the Dunn’s test at the 5% level of significance” below each figure. Hopefully, the correction has been much improved (Please see the red and blue color texts of the Figures at pages 5–9 of revised manuscript). Please see the revised manuscript and thanks for your great suggestion. We appreciate you giving us the second chance to revising the manuscript.
Yours truly,
Wen-Chieh Sung, Ph.D.
Professor
Department of Food Science
National Taiwan Ocean University

Reviewer 2 Report
Accept in the present form
Author Response
Dear Reviewer #2
The authors are extremely grateful to anonymous referee involved for providing his/her excellent comments and valuable advice in this paper. We have revised the paper based on the referee’s comments. We have pleasure in requesting the referee to review this paper. Thank you. Your prompt attention to this paper will be much appreciated.
Point 1: Accept in the present form.
Response 1: Please see the revised manuscript and thanks for the valuable comment.
Yours truly,
Wen-Chieh Sung, Ph.D.
Professor
Department of Food Science
National Taiwan Ocean University

Reviewer 3 Report
This manuscript has some interesting results and I judged "minor division".
My comment on this manuscript is below.
L37: Reference number should be revised from "[6-7]" to "[6,7]". The same applies below.
L92-93: The sentence on addition of chitosan should be modified. This process is not clear or difficult to understand.
L94: "the mixture was titrated with " What was determined by the titration?
L95: "the tubes with caps" What is contained in these tubes?
L112: "the acrylamide" should be changed to "acrylamide".
L114: Why is acrylamide washed with water? acrylamide is soluble in water.
I can't understand this process.
L123&L145: On calibration curve, If possible, can the authors show the molar extinction coefficient value, ε?
L172: "for 3 time" is "3 times"??
L196: "the formation of brown polymers" The explanation of these brown polymers is required.
L198: "Fig. 1 and 2" is changed to "Figs. 1 and 2".
L214: "the term "after 10 min of heating" is changed to "after heating for 10 min". The same applies below.
L218: "Fructose-asparagine dissolved in water" is changed to "aqueous Fructose-asparagine solution".
L226: "Adding chitosan to" is changed to "the addition of chitosan to".
L249: "HMF concentration" is changed to "the HMF concentration".
L261-268: What is shown (explained) from the change in viscosity?
L302: A sentence with the subject except person is better.
L352: The term "into" is not adequate. This sentence is revised.
L364: "Solutions containing" is changed to "The solutions containing".
L366: "a better understanding of MRPS" is changed to "a better understanding of (the) formation of MRPs".
References: Some underlines with doi should be deleted.
Author Response
Dear Reviewer #3
The authors are extremely grateful to anonymous referee involved for providing his/her excellent comments and valuable advice in this paper. We have revised the paper based on the referee’s comments. We have pleasure in requesting the referee to review this paper. Thank you. Your prompt attention to this paper will be much appreciated.
Point 1: This manuscript has some interesting results and I judged "minor division".
My comment on this manuscript is below.
L37: Reference number should be revised from "[6-7]" to "[6,7]". The same applies below.
Response 1:
We have rewritten the reference cited number through our article carefully as revised texts marked in red color in lines 38, 54, 58, 61, and 201 of the revised manuscript. Thanks for the suggestions (Please see the revised manuscript).
Point 2: L92-93: The sentence on addition of chitosan should be modified. This process is not clear or difficult to understand.
Response 2:
The sentence on the addition of chitosan was modified in the Materials and Methods section 2.2 of revised manuscript. Thanks for informing the problems in the 2.2 Preparation of Maillard reaction model system in the previous manuscript. Hopefully, the correction has been much improved (Please see the blue color texts at page 2 lines 90-101 of revised manuscript).
Point 3: L94: "the mixture was titrated with " What was determined by the titration?
Response 3:
Thank you for pointing out the sentence related to preparation of Maillard reaction model system of section 2.2. We have revised the word “titrated” to “adjusted” and added more description in the revised manuscript. Hopefully, the sentence has been much improved (Please see the red color texts of the section 2.2 line 95 at page 2 of revised manuscript).
Point 4: L95: "the tubes with caps" What is contained in these tubes?
Response 4:
We have rewritten the sentence of tubes by revising some texts at Materials and Methods section 2.2 at page 2 lines 90-101 of the revised manuscript. (Please see the revised manuscript).
Point 5: L112: "the acrylamide" should be changed to "acrylamide".
Response 5:
The phrase “the acrylamide” has been changed to “acrylamide” at page 3 line 115 of the revised manuscript as the marked red texts. Thanks for informing us the problem.
Point 6: L114: Why is acrylamide washed with water? Acrylamide is soluble in water. I can't understand this process.
Response 6:
We have revised and modified the reference (Roach et al., 2003) method information of acrylamide extraction and measurement at page 3 third paragraph as the marked red texts. The preconditioned Oasis HLB/MCX cartridge was used to absorb the acrylamide of the 3 mL filtrate. The cartridge was washed with 0.5 mL of deionized distilled water to clean up the filtrate and the other materials and the 0.5 mL eluate was discarded to avoid too much impurities for HPLC system. The sorbent acrylamide was washed with 3.0 mL of DD water. Thanks for the question and we appreciate you giving us the information which eluate might contain acrylamide.
Roach, J.A.G, Andrzejewski, D., Gay, M.L., Nortrup, D., Musser, S.M. (2003). Journal of Agricultural and Food Chemistry 51, 7547-7554
Point 7: L123&L145: On calibration curve, If possible, can the authors show the molar extinction coefficient value, ε?
Response 7:
The molar extinction coefficient (ε) is a measure of how strongly a chemical absorbs light at a particular wavelength. The molar extinction coefficient is frequently used in spectroscopy to measure the concentration of a chemical in solution. However, the concentration of acrylamide of the tested samples were determined using HPLC method. The acrylamide concentration was calculated from the integral area of UV absorbance converting to concentration by acrylamide standard solution not by how strongly acrylamide absorbs at 210 nm UV light of the manuscript.
Point 8: L172: "for 3 time" is "3 times"??
Response 8:
We are sorry for grammar mistake of Materials and Methods section. The information of the sentence in section 2.4 has been revised to 3 times. Thanks for informing us the mistake in the previous manuscript. (Please see the revised manuscript at page 4 line 177).
Point 9: L196: "the formation of brown polymers" The explanation of these brown polymers is required.
Response 9:
The term “the formation of brown polymers” was revised to “the formation of brown intensity of the Maillard reaction products”. Thanks for pointing out the miss meaning turn for us. (Please see the revised manuscript at page 5 line 201).
Point 10: L198: "Fig. 1 and 2" is changed to "Figs. 1 and 2".
Response 10:
The “Fig. 1 and 2” is revised to “Figs. 1 and 2”. Sorry for the mistake. (Please see the revised manuscript at page 5 line 198).
Point 11: L214: the term "after 10 min of heating" is changed to "after heating for 10 min". The same applies below.
Response 11:
All the similar terms “after 10 min of heat” in the article were revised to “after heat for 10 min” through the whole revised manuscript in lines 12, 15, 210, 228, 254, 329, 378, 388, 399, 406, 407, 417, 419, 434, 437, 459, 497, 502, 507, 524, 561, 564, 566, and 567 as marked red texts. (Please see the revised manuscript.)
Point 12: L218: "Fructose-asparagine dissolved in water" is changed to "aqueous Fructose-asparagine solution".
Response 12:
The phrase “Fructose-asparagine dissolved in water” is revised as page 6 section 3.2 of lines 224-239 of revised manuscript.
Point 13: L226: "Adding chitosan to" is changed to "the addition of chitosan to".
Response 13:
The term “Adding chitosan to” is revised as the revised manuscript at page 6 lines 224-239.
Point 14: L249: "HMF concentration" is changed to "the HMF concentration".
Response 14:
The term “HMF concentration” is revised to “The HMF concentration” in the revised manuscript. Thanks for pointing out the grammar problem in our paper again. (Please see the marked red text at line 265 of revised manuscript.)
Point 15: L261-268: What is shown (explained) from the change in viscosity?
Response 15:
The brief explanation of decreased viscosity was added to the end of the sentence and discussed at the discussion section showed that chitosan was hydrolyzed. Please see the revised manuscript at line 277 and the second paragraph of Discussion section 4.3 at page 14.
Point 16: L302: A sentence with the subject except person is better.
Response 16:
The sentence was revised without using person as the subject. Please see the lines 324–329 of revised manuscript.
Point 17: L352: The term "into" is not adequate. This sentence is revised.
Response 17:
The term “into” was deleted and revised as “The addition of heated 1% chitosan (50–190 kDa) enhanced the intensity of OD294 and OD420 of asparagine-fructose solution (AFC) [15]” at section 4.1 lines 378-380 of revised manuscript.
Point 18: L364: "Solutions containing" is changed to "The solutions containing".
Response 18:
The terms “Solutions containing" was revised to "The solutions containing". Please see the section 4.2 of revised manuscript at page 11 line 394.
Point 19: L366: "a better understanding of MRPS" is changed to "a better understanding of (the) formation of MRPs".
Response 19:
The term "a better understanding of MRPS" was revised to "a better understanding of the formation of MRPs". Please see line 396 at the section 4.2 of revised manuscript at page 11.
Point 20: References: Some underlines with doi should be deleted.
Response 20: All underlines with doi were removed. Thanks for reminding the problem in reference section. The authors appreciate to your valuable comments and advice in this paper again.
Yours truly,
Wen-Chieh Sung, Ph.D.
Professor
Department of Food Science
National Taiwan Ocean University

Round 2
Reviewer 1 Report
The revised form of paper titled (Effect of chitosan incorporation on the development of acrylamide during Maillard reaction in fructose–asparagine model solution and the functional characteristics of the resultants) was improved to a good extent but some corrections are still mandatory for presentation of correct stat analysis and hence conclusion of the paper>
1- The symbols on the columns in figures are not correct still.
2- Authors should mention in figure legends what the letters a,b,c,d, stands for????
WHat does the current situation means?
We usually use:
a: compare to the first left group
b: compared to second left group & so on
So in your case the max number of letteres is 3, how you use 4 symbols? this means you repeated a comparison
Author Response
Dear Reviewer #1
The authors are extremely grateful to anonymous referee involved for providing his/her excellent comments and valuable advice in this paper. We have revised the paper based on the referee’s comments. We have pleasure in requesting the referee to review this paper. Thank you. Your prompt attention to this paper will be much appreciated.
Comment 1: The revised form of paper titled (Effect of chitosan incorporation on the development of acrylamide during Maillard reaction in fructose–asparagine model solution and the functional characteristics of the resultants) was improved to a good extent but some corrections are still mandatory for presentation of correct stat analysis and hence conclusion of the paper.
1- The symbols on the columns in figures are not correct still.
2- Authors should mention in figure legends what the letters a,b,c,d, stands for????
What does the current situation means?
We usually use:
a: compare to the first left group
b: compared to second left group & so on
So in your case the max number of letters is 3, how you use 4 symbols? this means you repeated a comparison
Response comment 1:
Thank you for your valuable comment on our manuscript with a special concern on the assignment of symbol letters above the bars in our created figures.
As the data were not normally distributed, a non-parametric Kruskal-Wallis test was used to look for significant differences (p < 0.05) between the treatment groups, followed by a post hoc test. Dunn's post hoc tests with Bonferroni correction were carried out on each pair of groups. In our study, there were 12 groups (3 solution groups × 4 heating-time groups) in one figure (For example, Fig. 1A). All groups were treated as independent variables.
Next, we assigned the symbol letter following the method by Piepho [1] to describe the difference between the paired groups. We followed the wording recommendation provided by Piepho [2] to describe the symbol letter above the bars. As the paired groups were compared using Dunn’s test, thus, symbol letters above the bars represented the mean rank comparison between the compared groups. The assignment of symbol letters was not manually inserted in the graph but facilitated by R statistical program using the rcompanion package to prevent human errors. As result, bars followed by a common letter in a figure are not significantly different by the Dunn’s test at the 5% level of significance (please see the wording below each graph).
We hope this clarifies the reviewer’s question.
References
1 Piepho, H.P., 2004. An algorithm for a letter-based representation of all-pairwise comparisons. Journal of Computational and Graphical Statistics, 13(2), pp.456-466.
2 Piepho, H.P., 2018. Letters in mean comparisons: What they do and don’t mean. Agronomy Journal, 110(2), pp.431-434.
We have rewritten and added two sentences as revised texts marked in red color in the Conclusion section of revised manuscript. Thanks for the suggestions and we appreciate you giving us the second chance to revising the manuscript. (Please see pages 5-11 (Figures 1-8), 15, and 16 (Conclusions) of the revised manuscript).
Yours truly,
Wen-Chieh Sung, Ph.D.
Professor
Department of Food Science
National Taiwan Ocean University

This manuscript is a resubmission of an earlier submission. The following is a list of the peer review reports and author responses from that submission.
Round 1
Reviewer 1 Report
I have the following comments:
- The abstract should contain also the possible application of the experiment.
- The aim is not clearly defined at the end of the introduction part.
- The source of chitosan in food should be emphasized in the introduction part, especially because chitosan is used recently as the part of edible/biodegradable packaging and films. The following reference should be used: Jancikova, S., Dordevic, D., Tesikova, K., Antonic, B., & Tremlova, B. (2021). Active edible films fortified with natural extracts: Case study with fresh-cut apple pieces. Membranes, 11(9), 684.
- The analysis done by HPLC system should contain the exact information about used: mobile phase, detector, colon, flow, temperature and producer.
- It is not clear how asparagine was detected; what kind of device was used. Please write information in detail.
Author Response
Dear Reviewer #1
The authors are extremely grateful to anonymous referee involved for providing his/her excellent comments and valuable advice in this paper. We have revised the paper based on the referee’s comments. We have pleasure in requesting the referee to review this paper. Thank you. Your prompt attention to this paper will be much appreciated.
Point 1: The abstract should contain also the possible application of the experiment.
Response 1: We added the possible application of the experiment to the last sentence of abstract section as the texts marked in red color in the revised manuscript. Please see the abstract of revised manuscript and thanks for your great suggestion.
Point 2: The aim is not clearly defined at the end of the introduction part.
Response 2: We revised the last few sentences at the end of the introduction section as the texts marked in red color to define the aim of this research. Hopefully, it has a great improvement. We will revise it again if it is good enough at this time. Thanks for informing the problems in the Introduction section in the previous manuscript.
Point 3: The source of chitosan in food should be emphasized in the introduction part, especially because chitosan is used recently as the part of edible/biodegradable packaging and films. The following reference should be used: Jancikova, S., Dordevic, D., Tesikova, K., Antonic, B., & Tremlova, B. (2021). Active edible films fortified with natural extracts: Case study with fresh-cut apple pieces. Membranes, 11(9), 684.
Response 3:
Thanks for informing us the important related study. We already add and cite the new information to the introduction and reference sections. Please see the revised manuscript at pages 2 (lines 6 & 8) and 6 reference #12.
Point 4: The analysis done by HPLC system should contain the exact information about used: mobile phase, detector, column, flow, temperature and producer.
Response 4: We added more detail information for analysis acrylamide, HMF and asparagine conditions for HPLC methods including mobile phase, detector, column, flow rate, operating temperature and the brand name of HPLC producer. Please see the Materials and Methods section of the revised manuscript at page 3. Thanks again for the helpful suggestion.
Point 5: It is not clear how asparagine was detected; what kind of device was used. Please write information in detail.
Response 5: We used the same HPLC system but different wavelength for measuring the absorbance of asparagine derivative as the condition mentioned in the revised Materials and Methods of revised manuscript at pages 3 & 4.
Yours truly,
Wen-Chieh Sung, Ph.D.
Professor
Department of Food Science
National Taiwan Ocean University

Reviewer 2 Report
Paper titled (Effect of chitosan incorporation on the development of acrylamide during Maillard reaction in fructose–asparagine model solution and the functional characteristics of the resultants) by Lin et al. tested the effect of adding chitosan on the Millard reaction in a food model solution containing 0.5% fructose and asparagine after 30 min heating at 180°C. This study is of a good value. Although methods are well written, RESULTS are NOT have the following comments for improvemnet.
1- Authors need to check the normality of distribution of the data by a suitable test such as Shapiro-Wilk or K-S tests before deciding to use one-way ANOVA. Many of the data seem to be not normally distributed and deserve non parametric ANOVA and Dunn test
2- Figures are not acceptable: symbols are repeated and making redudence making the figures confusing
Authors should use symbols in order
a Compared to first column
b compared to second column & so on
So among 4 groups, 3 symbols are only needed, why all these symbols unti E
Do not come back by symbols comparing the left columns with the right ones
We walk from the left to the right.
Please consult a statistician and reformulate the results before we can read the conclusion from this study
Author Response
Reviewer 2’s comments
Comments and Suggestions for Authors
Paper titled (Effect of chitosan incorporation on the development of acrylamide during Maillard reaction in fructose–asparagine model solution and the functional characteristics of the resultants) by Lin et al. tested the effect of adding chitosan on the Millard reaction in a food model solution containing 0.5% fructose and asparagine after 30 min heating at 180°C. This study is of a good value. Although methods are well written, RESULTS are NOT have the following comments for improvemnet.
- Authors need to check the normality of distribution of the data by a suitable test such as Shapiro-Wilk or K-S tests before deciding to use one-way ANOVA. Many of the data seem to be not normally distributed and deserve non parametric ANOVA and Dunn test
Thank you very much for your comments.
We managed to use Shapiro-Wilk or K-S tests to run the analysis; however, the data do not seem to have a normal distribution. We further noticed that to use Shapiro-Wilk or K-S tests for statistical analysis might be suitable for some specific data, such as age distribution or body weight distribution in a group.
We would like to mention that using analysis of variance (ANOVA) and Duncan’s test for statistical analysis is a common approach to analyze Bioscience data. In addition, the main conclusion from this study is to indicate that addition of chitosan decreased in the acrylamide content during 30 min of heating in a fructose–asparagine system and that the formation of ultraviolet-absorbing intermediates and browning intensity of MRPs containing acrylamide prepared by fructose–asparagine was more than those of MRPs prepared by glucose–asparagine solution system. We would suggest that these results support the conclusion.
2- Figures are not acceptable: symbols are repeated and making redudence making the figures confusing
Authors should use symbols in order
a Compared to first column
b compared to second column & so on
So among 4 groups, 3 symbols are only needed, why all these symbols unti E
Do not come back by symbols comparing the left columns with the right ones
We walk from the left to the right.
Please consult a statistician and reformulate the results before we can read the conclusion from this study
Thank you very much for your comments.
Statistical analysis was used to analyze and compare every piece of data in the figure, and the lowercase letters are used to report the results of all the comparisons among all groups and treatments (lowercase “a” denoted the group with the largest number of all). As a results, the numbers of different lowercase letters in the figure represent the numbers of variables in statistical analysis.

Round 2
Reviewer 1 Report
Greetings,
please add information about the preparation of samples for the HMF determination.
Asparagine detection is not clear, what kind of detector was used, what was mobile phase?
The manuscript still need more information about used methods.
Reviewer 2 Report
The revised form of the article was very partly improved, essential issues were not resolved. Authors did not correct results although they confirmed they are not in normal distribution. They had to manage differences by using non parametric ANOVA instead of one-way ANOVA...etc.
Also the second comment regarding the diff sympols on the column charts were not corrected. Figure legends are very confusing and non-understandable. a-i: means sig diff????!!!! from what and how???
Authors are in a need to consult a statistician for how to present their data.